# Proteome-Level Investigation of *Vitis amurensis* Calli Transformed with a Constitutively Active, Ca^2+^-Independent Form of the *Arabidopsis AtCPK1* Gene

**DOI:** 10.3390/ijms241713184

**Published:** 2023-08-24

**Authors:** Galina N. Veremeichik, Dmitry V. Bulgakov, Yuliya A. Konnova, Evgenia V. Brodovskaya, Valeria P. Grigorchuk, Victor P. Bulgakov

**Affiliations:** Federal Scientific Centre of the East Asia Terrestrial Biodiversity of the Far East Branch of the Russian Academy of Sciences, 690022 Vladivostok, Russia; bulgakov-dv@mail.ru (D.V.B.); kera1313@mail.ru (V.P.G.); bulgakov@biosoil.ru (V.P.B.)

**Keywords:** *Arabidopsis thaliana*, *Vitis amurensis*, calcium-dependent protein kinases, MALDI-TOF, plant defense, pathogenesis related protein, proteomics

## Abstract

Calcium-dependent protein kinases (CDPKs) are one of the main Ca^2+^ decoders in plants. Among them, *Arabidopsis thaliana AtCPK1* is one of the most studied CDPK genes as a positive regulator of plant responses to biotic and abiotic stress. The mutated form of *AtCPK1*, in which the autoinhibitory domain is inactivated (*AtCPK1-Ca*), provides constitutive kinase activity by mimicking a stress-induced increase in the Ca^2+^ flux. In the present study, we performed a proteomic analysis of *Vitis amurensis* calli overexpressing the *AtCPK1-Ca* form using untransformed calli as a control. In our previous studies, we have shown that the overexpression of this mutant form leads to the activation of secondary metabolism in plant cell cultures, including an increase in resveratrol biosynthesis in *V. amurensis* cell cultures. We analyzed upregulated and downregulated proteins in control and transgenic callus cultures using two-dimensional gel electrophoresis, and Matrix-assisted laser desorption and ionization time-of-flight mass spectrometry (MALDI-TOF). In calli transformed with *AtCPK1-Ca*, an increased amounts of pathogenesis-related proteins were found. A quantitative real-time PCR analysis confirmed this result.

## 1. Introduction

Calcium is the most important second messenger coordinating the physiological response of plants to external or internal signals. An adequate response is provided by the “calcium signature”. Various signals, including stress signals, cause a unique combination of intracellular calcium fluctuations [1]. In plants, the function of decoding calcium signals is performed by four classes of enzymes. One of them is calcium-dependent protein kinases, or CDPKs. The uniqueness of these enzymes lies in their versatility: CDPKs are simultaneously able to bind directly to calcium, bypassing any intermediaries, and can activate certain target proteins due to substrate-specific phosphorylation. Several dozen different isoforms of the CDPK genes are present in the plant genome. They differ from each other in parameters such as sensitivity to calcium, substrate specificity, localization, and many others. This diversity ensures the involvement of CDPKs in many cellular processes [2,3].

In plants, calcium-dependent protein kinases are the main transducers of an external negative signal that causes the acceleration of intracellular calcium. CDPK transduces this signal through phosphocode-defined “logic gates” that dictate transcriptional reprogramming during protection [4]. For cellular response, CDPKs modulate the activity of key regulatory enzymes in major plant cell signaling systems.

The AtCPK1 isoform of Arabidopsis (GenBank accession number, AT5G04870; http://www.uniprot.org/uniprot/Q06850, accessed on 1 February 1995) is one of the most studied [5]. As a rule, CDPK proteins consist of four domains: N-terminal, C-terminal Ca^2+^ binding, kinase, and autoinhibitory domains. The N-terminal variable domain is involved in substrate specificity, while the C-terminal Ca^2+^ binding domain (CaM-like domain, CLD) activates the PK kinase domain [6,7]. CLD is associated with the PK domain through an autoinhibitory 35 a.a. domain (J) [3]. The J and CLD domains do not function independently [8] and form a single functional CDPK activation domain (CAD). The junction fragment blocks kinase activity when [Ca^2+^]_cyt_ is low. An increase in [Ca^2+^]_cyt_ in response to external influences leads to Ca^2+^ binding by the CAD, and conformational changes remove the J fragment from the protein kinase active site. Free PK domains provide access to the substrates [5]. Thus, manipulations with the CAD and J-fragment are the most promising for the genetic activation of CDPK. Genetic deletion of the CAD from *Arabidopsis* CPK1 or CDPKα in soy resulted in the activation of these kinases [9,10]. At the same time, the absence of the CAD in another CPK, CPK17 from rice (the AtCPK1 orthologue), resulted in a loss of catalytic in vitro activity [11]. Moreover, the inhibition of *Arabidopsis* CPK10 by EGTA indicated that the Ca^2+^-associated CAD may be important for the CPK function [9,12,13]. These are pioneering and interesting works, in which several types of point mutations in J of the *AtCPK1* gene and truncated mutants were analyzed. Among the point and truncated mutants, the KJM23 mutant was the most active in the presence or absence of Ca^2+^ (the mutation replaced AV-424 with PD and QFSA-430 with PEDL). The KJM4 mutation (a substitution of LRV-I444 to DLPG) has been described by Huang et al. [13]. This mutation disrupted Ca^2+^-induced activation, resulting in an inactive KJM4 [13].

KJM23 had a six-amino acid substitution between residues A422 and A432. This mutated form was highly active compared to other mutant forms, indicating that partial auto-inhibition could be obtained from non-homologous sequences located at the junction [9]. The KJM4 mutation in the junction domain completely abolished calmodulin binding. The C-terminus of the junction contains several overlapping sites (e.g., F436-I444 and F430-I444) that are similar to calmodulin binding domains [9,13]. Interestingly, both of the described substitutions alter the highly conserved amino acid residues of the CDPK junction domain. As described earlier, 35 a.a. in the junction domain contains the pseudosubstrate autoinhibitory subdomain (17 to 26 a.a.) and the CAM-BD (23 to 35 a.a.) subdomain [14]. Four of the six amino acid residues of the KJM23 mutation replace the most conserved residues of the pseudosubstrate autoinhibitory subdomain (QFSA to PEDL, Appendix A). The KJM4 mutation replaces the most conserved residues of the CAM-BD subdomain (LRVI to DLPG, Appendix A).

The effect of overexpression of these two mutant forms of the *AtCPK1* gene on plant cells was analyzed in comparison with the native form of *AtCPK1*. In our previous work, KJM23 and KJM4 were designated as *AtCPK1*-Ca (constitutively active) and *AtCPK1*-Na (inactive), respectively. It has been shown that *AtCPK1*-Ca has a strong activating effect on the biosynthesis of secondary metabolites in transformed cell cultures of *Rubia cordifolia* L., *Vitis amurensis* Rupr., and *Glycine max* L. [15,16,17]. In contrast, the cultures transformed with *AtCPK1*-Na did not have this ability [15,16,18].

The biosynthesis of secondary metabolites is a plant response to both biotic and abiotic stresses. Previously, *AtCPK1* was shown to positively regulate the response of plants to a pathogen attack, which is consistent with data on the role of CDPK in the regulation of secondary metabolism [19]. A transcriptome analysis of *CPK1*-overexpressed *Arabidopsis* plants showed that AtCPK1 activates the salicylic acid (SA) signaling system [19]. It has been shown in our experiments [16] that the overexpression of the constitutively active form of *AtCPK1* in *V. amurensis* cell cultures increases resveratrol biosynthesis without growth inhibition. However, there are currently no proteomic studies of plants with constitutive active forms of CDPK.

In the present study, we investigated changes in protein expression in an *AtCPK1-Ca*-transformed *V. amurensis* callus culture. Calli transformed with *AtCPK1-Ca* were shown to mimic a state similar to a stable stress-induced increase in Ca^2+^. The obtained data also showed the possible mechanism of action of *AtCPK1* in short-term stress signaling and the possibility of cell adaptation to prolonged stress.

## 2. Results

### 2.1. Matrix-Assisted Laser Desorption and Ionization Time-of-Flight Mass Spectrometric Proteomic

Using two-dimensional gel electrophoresis, total protein fractions from the control (non-transformed) and *AtCPK1-Ca*-transformed callus cultures of *V. amurensis* (Appendix A) were separated (Appendix A). An amount of 150 out of 1000 proteins separated on 2-D gels were identified using MALDI mass spectrometry. From them, 103 proteins were quantified. In total, 34 downregulated proteins and 38 upregulated proteins were identified in the *AtCPK1-Ca*-transformed callus culture. Using UniProtKB and TAIR databases, these proteins were grouped according to their function. Protein functional groups are represented by the following categories (the total protein number/upregulated/downregulated proteins): signaling (17/9/3), the DNA/RNA metabolic process (3/-/-), protein and amino acid synthesis (11/1/5), protein and amino acid catabolism (7/1/1), carbohydrate metabolic pathways (25/11/13), ATP synthesis (5/2/1), pathogenesis-related proteins (15/9/5), chaperones (7/3/3), the microtubule-based process (6/-/-), membrane transport (2/2/-), and other metabolic process (5/-/3). The mass spectrometry proteomics data have been deposited to the ProteomeXchange Consortium via the PRIDE [20] partner repository with the dataset identifier PXD043507 and 10.6019/PXD043507. Obtained data with protein names and functional characteristics are presented in Table 1 and Appendix A.

### 2.2. Plant Signaling Systems

We started our analysis with proteins involved in the signaling processes.

*Ethylene biosynthesis.* Among the identified proteins, we found three that are related to ethylene biosynthesis: S-adenosylmethioninesynthase (METK4_1) and two isoforms of bifunctional nitrilase/nitrilehydratases (NIT4B). METK is the main enzyme in ethylene biosynthesis. METK4_1 catalyzes the general precursor of ethylene, S-adenosylmethionine (SAM), from methionine [21]. NIT4B plays a key role in the catabolism of toxic nitriles, converting β-cyanoalanine to asparagine and aspartic acid. β-cyanoalanine is a detoxified derevative of hydrogen cyanide, a byproduct of ethylene biosynthesis [22,23]. The expression of METK4_1 was equal in the control Va and transgenic VaCa calli, while the abundance of two isoforms of NIT4B was increased in VaCa calli by two and eight times (Table 1, Figure 1a).

*Calcium signaling*. Three proteins of the calcium signaling system were identified: two isoforms of calreticulin (CRT) and annexin (ANN). Annexins are multifunctional proteins involved in actin modeling and membrane dynamics, [Ca^2+^]_cyt_ and ROS regulation [24]. CRTs are multifunctional proteins of the endoplasmic reticulum (ER) with a high affinity for Ca^2+^ and are well known as Ca^2+^-binding molecular chaperones. CRTs facilitate the folding of newly synthesized glycoproteins and regulate Ca^2+^ homeostasis in the ER lumen. Both isoforms of plant CRTs are involved in biotrophic pathogen defense regulation [25]. Our data showed an eight-fold increase in CRT expression and a five-fold decrease in ANN expression in the VaCa calli (Table 1, Figure 1b,c).

*Auxin, brassinosteroid, and MeJA signaling*. In the present study, we discovered several proteins associated with auxin, brassinosteroid, and MeJA signaling. 2-oxoglutarate-dependent dioxygenase (DAO) plays an important role in the auxin catabolic process and intracellular auxin homeostasis. Its expression was more than five times higher in the VaCa calli. Protein EXORDIUM (EXO) and peptidyl-prolyl cis-trans-isomerase are involved in brassinosteroid signaling; expression of these proteins is strongly decreased in the VaCa calli (Table 1, Figure 1d). The protein TIFY10c, a homolog of AtJAZ1 [26], was the sole identified member of MeJA signaling. Its expression was equal in the control and VaCa cell lines.

ROS and programmed cell death (PCD) signaling. PCD is directly connected to ROS signaling [27]. We have found a strong increase, more than seven times, in the expression of the PCD-related epidermis-specific secreted glycoprotein (EP1) in the VaCa calli. We have identified six ROS-scavenging proteins: three isoforms of glutathione S-transferase (GST), two isoforms of catalases (CAT), monodehydroascorbate reductase (MDAR), and class III peroxidase (PNC1_16). Among them, the expression of GSTs was increased 2–7 times, while expression of PNC1_16 was strongly reduced in VaCa cells (Table 1).

### 2.3. DNA/RNA, Amino Acid, and Protein Metabolic Processes

Three proteins related to the DNA/RNA biosynthetic process with equal expression in the control Va and transgenic VaCa cell lines were identified: adenosine kinase (ADK2), alpha mitochondrial processing peptidase (Alpha-MPP), and nucleoside diphosphate kinase (NPK). Several important proteins were found in the group related to amino acid and protein biosynthesis. Among 11 identified proteins, the expressions of five were equal in the control Va and transgenic VaCa calli. Expression levels of two proteins were increased five-fold in the transgenic VaCa calli, such as the 60S acidic ribosomal protein P0 (RLA0_4) and cysteine synthase (CYSK_6). The expression of elongation factor 1-alpha (TEF1_1), glutamine synthetase (GS), and phosphoserine aminotransferase (PAT) was strongly (8–9 times) decreased in the transgenic VaCa calli. The expression of ketol-acid reductoisomerase KAR and glutamated dehydrogenase GDH1_1 was slightly (4–6 times) reduced in the transgenic VaCa calli (Table 1).

The next considered group was presented by proteins related to the catabolism of proteins and amino acids. In this group, seven proteins were identified. Five of them were equal in the control Va and transgenic VaCa calli. Cyclase (Cyc), which regulates tryptophan catabolic processes, was slightly overexpressed in the transgenic VaCa calli, while the aspartyl protease AED3 related to senescence was abolished in the transgenic VaCa calli, and another putative cysteine protease, RD21B, was significantly increased (Table 1, Figure 2a,b).

### 2.4. Carbohydrate Metabolic Pathways and ATP Synthesis

We have identified 25 proteins related to carbohydrate metabolic pathways. From them, only two isoforms of phosphopyruvate hydratase (ENO) were equal in the control Va and transgenic VaCa calli. Expression levels of 11 enzymes were increased (3–9-fold), while the expression of 13 enzymes was slightly or significantly suppressed (Table 1). It was found that enzymes involved in the biosynthesis of triose phosphates and glycolysis (such as UTP-glucose-1-phosphateuridylyl transferase and fructose-bisphosphate aldolase) were strongly downregulated in VaCa calli, while enzymes involved in the penthose phosphate cycle were upregulated (such as glyceraldehyde-3-phosphate dehydrogenase and phosphoglyceratekinase). Moreover, among enzymes involved in the tricarboxylic acid cycle; enzymes of L-ascorbic acid biosynthesis, isocitrate metabolism, and processing; and fatty acid biosynthesis, several proteins were downregulated (such as malate dehydrogenase, aconitate hydratase, and isocitrate dehydrogenase). At the same time, the expression of the main enzyme involved in mitochondrial mROS production, malic enzyme (NADP-ME), was significantly (more than eight times) increased (Table 1, Figure 2c); the abundance of succinate-CoA ligase and alcohol dehydrogenase 1 was also significantly increased in the transgenic VaCa.

ATP biosynthesis, like carbohydrate biosynthesis, is a common part of energy metabolism. We identified five proteins related to ATP synthesis. The levels of expression of two of them were equal in the control Va and transgenic VaCa calli: the vacuolar proton pump subunit (VPP) and V-ATPase 69 kDa subunit (VATA_3). The expression of the ATP synthase subunit alpha (ATPA) was decreased by more than seven times, while the expression of two isoforms of the ATP synthase subunit beta (ATPB) was increased by 3–4 times in the transgenic VaCa calli (Table 1).

### 2.5. Pathogenesis-Related Proteins

Overall, 15 pathogenesis-related (PR) proteins were identified in this work. The expression of only one of them (endochitinase, EP3_15) was equal in the control Va and transgenic VaCa calli. Other proteins were strongly upregulated (eight proteins) or strongly downregulated (seven proteins) in the transgenic VaCa calli. Among the eight upregulated proteins, three proteins were of a low abundance (almost undetectable) in the control cells: glucanendo-1,3-beta-d-glucosidase (BG3), betvI/major latex protein domain-containing protein (PR-10; Table 1, Figure 3a), and pathogenesis-related protein 10.3 (PR10.3). Two isoforms of PR proteins (PR10.1 and PR10.2) were strongly upregulated, more than nine times, in the transgenic VaCa calli. Interestingly, besides the PR proteins from group 10, we found that the expression of two PR5-like proteins, osmotin-like 13 (Ol13_1) and thaumatin-like 3 (Tl3), was significantly increased, more than five times, in the transgenic VaCa calli (Table 1, Figure 3b,c).

In transgenic VaCa calli, the expression of two isoforms of endoglucanase (CEL1_4 and GUN_4) was decreased (3–4 times). The expression of three PR proteins such as beta-D-xylosidase 1 (BXL1), class IV chitinase (Chi4D), and glucanendo-1,3-beta-D-glucosidase (E13ip5.23) was completely suppressed (Table 1, Figure 3d,e). The expression of their homologs, beta-xylosidase/alpha-L-arabinofuranosidase 2 (Xyl2_0) and glucanendo-1,3-beta-D-glucosidase (HGN1_0), was strongly reduced but not abolished.

### 2.6. Chaperones

In the present investigation, we identified seven proteins that belong to the chaperone group. First, it is the RuBisCO large subunit-binding protein subunit alpha (RUB1_1) and subunit beta (RUBB_0). The expression of the RUB1_1 was equal in the control Va and transgenic VaCa calli, while expression of the RUBB_0 was moderately (four times) increased in the transgenic VaCa calli. Expression of the histidine kinase/HSP90-like ATPase domain-containing protein (HSP90) and mitochondrial chaperonin CPN60-2 (CPN60-2_1) was significantly increased (5–7 times) in the transgenic VaCa calli. We also detected three isoforms of HSP70. One of them, the mitochondrial heat shock protein (HSP7M_1), was expressed equally, while the expressions of HSP70_18 and HSP70_16 were strongly decreased in the transgenic VaCa calli (Table 1, Figure 4a,b).

### 2.7. Other Process

Other analyzed proteins include proteins involved in microtubule-based processes, membrane transport, and other metabolic processes. We have identified eight proteins involved in the microtubule-based process: five isoforms of tubulin and three isoforms of actin; however, no significant changes in the abundance of these proteins were found (Table 1). Interesting results were obtained for proteins associated with membrane transport and the process of salicylic acid biosynthesis. In one spot, we found a complex of two proteins: mitochondrial outer membrane protein porin (34 kDa) and prohibitin. The expression of these proteins was significantly increased in the transgenic VaCa calli, more than five times. Besides this, we identified five proteins that belong to different metabolic processes. From them, three proteins were strongly downregulated in the transgenic VaCa (GDSL esterase/lipase, monooxygenase 3, and pectin esterase (Table 1).

### 2.8. qRT-PCR Analysis of Some Important Proteins

To validate proteomic data, we performed a qRT-PCR analysis of some important proteins. In addition to the control Va and *AtCPK1*-*Ca*-transformed callus lines, we used VaNa and VaCa lines [16]. The VaNa line expressed an inactive form of *AtCPK1*, called *AtCPK1-Na*. The VaCa’ line expressed the *AtCPK1*-*Ca* gene at a lower level than the VaCa line (Appendix A). We performed three independent RNA extractions and analyzed the expression of genes of the most important *AtCPK1-Ca*-induced proteins: chaperones (*HSP90*); malic enzyme (*ME*); bifunctional nitrilase/nitrilehydratase (*NIT*); cysteine protease (RD21); PR5, thaumatin-like (*Tl*) and osmotin-like (*Ol*) proteins; and *PR10*, the betvI/major latex protein (*Bet*). Representative results of the qRT-PCR analysis are presented in Figure 5. Relative changes in expression were compared to the control Va calli and calculated as 2^−ΔΔCt^. The data are presented in different colors: pink, VaNa; blue, VaCa; and green, VaCa’. The qRT-PCR data correlated with the accumulation of some proteins in VaCa cells, which may be the result of this induced transcription. Interestingly, the expression of *RD21* protease was similar in VaCa and VaCa’ calli, while the expression of other genes was more than two times higher in the high *AtCPK1-Ca*-expressed VaCa line.

## 3. Discussion

MALDI-TOF MS (Matrix-assisted laser desorption and ionization time-of-flight mass spectrometry) is a powerful technique in plant proteomics [28]. The techniques of 2-D gel electrophoresis and MALDI mass spectrometry have been extensively used in basic research to address the question of how plants perceive and transduce endogenous and environmental signals [28]. Presently, callus cultures are widely used in proteomic experiments as a universal model, providing a relatively standardized and homogeneous basis for research. Plant callus cultures were successfully used in proteomic experiments to study the molecular mechanisms underlying different aspects of cell differentiation and somatic embryogenesis, stress adaptation, and *Agrobacterium*-plant interaction [29,30,31,32,33,34]. In the present work, we analyzed *V. amurensis* callus cultures with an overexpression of *CDPK* by 2-D gel electrophoresis and MALDI mass spectroscopy. The analysis of the overexpression of *CDPK* in plant cells was performed for the first time. Previous studies have shown the involvement of *AtCPK1* in processes such as the biosynthesis of secondary metabolites, resistance to abiotic [35,36] and biotic [19] stresses, as well as the effect on ROS metabolism [35,36]. However, it remained unclear how these processes could be related. The present study was undertaken to create a working hypothesis for the functioning of this kinase, for which a permanently active form was used.

Based on the results of the MALDI analysis, we propose at least three potential CDPK-related targets for future detailed investigations. First, *AtCPK1*-induced changes in the composition of PR proteins were found. In VaCa cells, the expression of PR-2 (glucanendo-1,3-beta-d-glucosidase) and PR-5 (osmotin-like and thaumatin-like proteins) proteins increased, while the expression of PR-3 (class IV chitinase) decreased. It is known that PR-2 and PR-5 are proteins expressed in association with SAR (systemic acquired resistance), while PR-3 is expressed during LAR (local acquired resistance). This may be caused by the induction of SA signaling and the inhibition of JA signaling [37,38,39]. In the VaCa culture, the expression of three forms of PR-10 proteins (PR10.1, PR10.2, and PR10.3) was significantly (9–10 times) increased. The increase in the amount of PR-10 proteins in the VaCa culture is consistent with previous studies, because it was shown that the expression of PR-10 proteins is regulated through a JA-independent pathway [19]. Changes in PR proteins correlate well with previous data on the induction of secondary metabolite biosynthesis upon the overexpression of the *AtCPK1* gene and indicate an increase in the overall defense status of cells. Interestingly, these changes were not accompanied by a decrease in the biomass of cell cultures [15,16,17].

Secondly, there were changes in the primary metabolism. MALDI mass spectrometry revealed an interesting aspect of proteome changes in the pathways of primary metabolism. To discuss these results, we visualized the proteins involved in primary metabolism, sugar metabolism, and amino acid biosynthesis in one general scheme (Figure 6). A strong increase in the abundance of enzymes involved in glyceraldehyde-3-phosphate (G3P) biosynthesis seems to point to the activation of the biosynthesis of primary precursors of the ABA and brassinosteroid biosynthetic pathways and the precursors of the general products of the aromatic amino acid family (AAF): SA from chorismate, IAA from tryptophan, and phenylpropanoids, including stilbenes from phenylalanine and tyrosine. Interestingly, only one enzyme group from the TCA cycle is activated in the VaCa cells: malic enzymes from the malate cycle. This may lead to the regeneration of NADPH as a reluctant for other anabolic processes and the production of mitochondrial ROS (mROS), as indicated by Zhao et al. [40]. One of the mROS scavengers is the prohibitin/porin complex [41]. As shown earlier, prohibitin forms complexes with ICS to activate stress-induced SA biosynthesis [42]. In our study, a significant increase in prohibitin expression in VaCa cells was shown.

The third change, not previously noted in our work or in the work of other authors, is a change in the expression of proteins that regulate hormonal metabolism. The strong *AtCPK1-Ca*-mediated activation of phosphoglycerate biosynthesis and the biosynthesis of cysteine from serine can be assessed as a change in the biosynthesis of methionine. While the biosynthesis of acetyl-CoA is activated, other stages of the tricarboxylic acid (TCA) cycle are strongly inhibited; therefore, the biosynthesis of glutamate from citrate and aspartate from oxaloacetate could be blocked. As a result, not only the biosynthesis of glutamate and γ-aminobutyric acid (GABA) but also the biosynthesis of fatty acids could be blocked. It is possible that the inhibition of aspartate biosynthesis leads to the weakening of another biosynthetic pathway, namely the methionine biosynthesis pathway as a precursor of ethylene. However, an alternative methionine biosynthetic pathway is strongly activated. Due to the fact that the bifunctional nitrilase/nitrile hydratase, the detoxification enzyme of the cyanide by-product of ethylene, is strongly activated, it can be assumed that ethylene biosynthesis can also be activated. This result is in line with previous research indicating that CDPK activates ethylene biosynthesis under stress conditions through the phosphorylation of biosynthesis enzymes [44].

It seems likely that an AtCPK-Ca-induced alteration in the biosynthesis of signaling molecules reprogrammed the immune status to SAR (Figure 7). The inhibition of some key enzymes involved in GABA biosynthesis pointed to the important role of GABA in the immune status [45]. It is known that GABA plays a role in the regulation of ABA and ethylene signaling [45]. The role of brassinosteroids (BS) in this process may also be important. We did not obtain data indicating changes in BS biosynthesis. However, some regulatory elements of the BS pathway were changed. For example, the content of EXORDIUM, which is an important mediator of BS signaling [46], was significantly reduced in VaCa cells, and the expression of some BS-dependent HSPs was reduced (Table 1), in accordance with previous data [47]. There are many hypothetical conclusions in these results, but a working scheme for in-depth experiments can already be created. With regard to ethylene signaling, its role in crossing various signaling pathways is already well documented. Along with BS, SA, and JA, ethylene plays an important role in the mutual regulation of the biotic stress response [48].

## 4. Materials and Methods

### 4.1. Plant Material

The control untransformed calli of *V. amurensis* Rup (*Vitaceae*) and *AtCPK1*-transformed calli were cultivated in 100 mL Erlenmeyer flasks on a WB/A medium supplemented with 0.5 mg L^−1^ 6-benzylaminopurine and 2.0 mg L^−1^ α-naphthaleneacetic in the dark at 25 °C with 30-day subculture intervals as described previously [16]. For the proteomics analysis, we used a control untransformed line Va and *AtCPK1*-Ca transformed line VaCa with a high transgene expression. For the qPCR analysis, we additionally used *AtCPK1*-Na transformed calli VaNa and an *AtCPK1*-Ca transformed line VaCa’ with a low transgene expression. All parameters of these calli lines, such as their obtaining, growth indexes, biochemical characteristics, and transgene expression, were described previously [16]. For the proteomic analysis, we used 30-day-old calli in the stationary growth phase.

### 4.2. 2-D-Gel Electrophoresis

Reagents were purchased from Sigma-Aldrich (St. Louis, MO, USA), unless otherwise noted. Proteins were isolated from the fresh biomass of Va and VaCa calli (1 g fresh weight) using a SDS extraction buffer and a cold 20% TCA/acetone precipitation method. Protein quantification was performed using the Bradford assay. For isoelectric focusing, dried protein pellets were dissolved in the IPG buffer, containing 9.5 M urea with thiourea, 4% *w*/*v* CHAPS, 65 mM DTT, 2% Pharmalyte pH 3–10 (GE Healthcare, Uppsala, Sweden), and 0.01% *w*/*v* bromophenol blue. A protein probe diluted in the IPG buffer was loaded to a 11-cm Immobiline DryStrip pH 3–10 NL (GE Healthcare, Uppsala, Sweden) according to the manufacturer’s recommendations by passive rehydration for 12 h at 20 °C. IEF was performed in a Protean IEF Cell (Bio-Rad Laboratories Inc., Hercules, CA, USA) for 60,000 V-h. For SDS-PAGE, 12% polyacrylamide gels with 4% stacking gels were run in a Protean II xi cell (Bio-Rad Laboratories Inc., Hercules, CA, USA). The gels were stained with Coomassie Brilliant Blue G-250. A set of three gels for Va and VaCa was used in the analysis.

### 4.3. Quantification of Protein Expression

Gels were scanned using the VersaDoc MP 4000 System (Bio-Rad Laboratories Inc., Hercules, CA, USA). PDQuest 8.0.1 Advanced software (Bio-Rad Laboratories Inc., Hercules, CA, USA) was used for the analysis of the protein maps. The Spot Detection Wizard was used to select the parameters for spot detection, such as a faint spot and a large spot cluster. The results of automated spot detection were checked and manually corrected. A local regression model (Loess) was used for the normalization of spot intensity. The protein expression was accessed using PDQuest 8.0.1 Advanced software and was presented as the mean total intensity of a defined spot in a replicate gel group. The spot quantity is the sum of the intensities of pixels inside the boundary. The fold of the protein expression change was accessed based on the mean protein intensity. For quantitative differentiation, a 1.5-fold change or higher in the average spot intensity was regarded as significant. The statistical significance of differences was assessed using the student’s t-test at a significance level of 0.05 in three replicates.

### 4.4. Experimental Design and Statistical Rationale

Three biological experiments were carried out with three technical replicates. The total number of samples analyzed by MALDI was 150. The number of technical replicates for protein identification by MALDI mass spectrometry was 2–3 (up to 5 for important and low-abundance proteins). Individual protein spots, selected on the basis of the image-analysis output, were excised and digested in-gel with trypsin (Trypsin V511, Promega, Madison, WI, USA), as previously described [49]. For MALDI-TOF identification, 0.5–1 µL of the sample (50% solution of acetonitrile in water, 0.1% TFA) was placed on a ground steel MALDI target plate, AnchorChip, or SmallAnchor (depending on the protein quantity), and 0.5–1 µL of the matrix (α-cyano-4-hydroxycinnamic acid) (Bruker Daltonics, Bremen, Germany) was added.

### 4.5. MALDI-TOF Mass Spectrometry and Protein Identification

All mass spectra were acquired with an Autoflex (Bruker Daltonics, Bremen, Germany) MALDI-TOF mass spectrometer with a nitrogen laser operated in the positive reflector mode (standard method RP 700–3500 Da.par) under the control of FlexControl software (version 3.4; Bruker Daltonics, Bremen, Germany). The analysis was performed in automatic mode (AutoXecute—automatic Run). The spectra were externally calibrated using the Calibrate Peptide Standards FAMS Method and a standard calibration mixture (Protein Calibration Standard I, Bruker Daltonics, Bremen, Germany). The data files were transferred to Flexanalysis software version 3.4 (Bruker Daltonics, Bremen, Germany) for automated peak extraction. The assignment of the first monoisotopic signals in the spectra was performed automatically using the signal detection algorithm SNAP (Bruker Daltonics, Bremen, Germany). For MS and MS/MS analyses, we used the PMF.FAMS Method and the SNAP_full_process FALIFT Method, respectively. Each spectrum was obtained by averaging 1500–5000 laser shots (300 shots in a step) acquired at the minimum laser power. The data was analyzed using BioTools (version 3.2; Bruker Daltonics, Bremen, Germany). A peptide mass tolerance of 0.5 Da and a fragment mass tolerance of 0.5 Da were adopted for database searches. The *m*/*z* spectra were searched against the *Arabidopsis thaliana* NCBI and SwissProt databases using the Mascot search engine. The threshold score was 40. Further data were analyzed using UniProtKB (http://www.uniprot.org/uniprot/, accessed on 12 June 2022) and other specialized databases and programs. The mass spectrometry proteomics data has been deposited to the ProteomeXchange Consortium via the PRIDE [20] partner repository with the dataset identifier: project name: Function of constitutively active AtCPK1; project accession: PXD043507; project DOI: 10.6019/PXD043507.

### 4.6. RNA Isolation, cDNA Synthesis, and Real-Time PCR

The total RNA isolation and characterization, as well as the synthesis of the first-strand cDNA, were performed as described previously [50]. RNA samples were isolated from the same samples used for protein extraction. The quantitative real-time PCR (qPCR) analysis was performed using an CFX96 (Bio-Rad Laboratories, Inc., Hercules, CA, USA) as described [51]. Three biological replicates, resulting from different RNA extractions, were used for the analysis, and three technical replicates were analyzed for each biological replicate. The gene-specific primer pairs used in the qPCR are listed in Appendix A. Housekeeping genes were described previously [16]. The lower-expressing sample was assigned the value 1 in the relative mRNA calculation using the formula 2^−ΔΔCt^. Data were analyzed using CFX Manager Software (Version 1.5; Bio-Rad Laboratories, Inc.).

### 4.7. Statistical Analysis

Statistica 10.0 (StatSoft Inc., Tulsa, OK, USA) was used for performing statistical tests. For comparison among multiple data, the analysis of variance (ANOVA) based in the Fisher’s protected least significant difference (PLSD) *post-hoc* test was employed for the inter-group comparison. Two independent categories were compared using the student’s *t*-test. A difference of *p* < 0.05 was considered significant.

## 5. Conclusions

Thus, the data obtained indicate the importance of further studies on the effect of CDPK at all three molecular levels: primary metabolism, the biosynthesis of signaling molecules, and PR expression. It is possible that the role of CDPK in the regulation of the plant cell response to stress is much wider than just the regulation due to the phosphorylation of components of phytoalexin biosynthesis pathways and ROS generation, but also the affect on other biochemical processes. The stable growth of *AtCPK1-Ca*-transformed calli [15,16,17,18] indicates that under normal growth conditions, a certain homeostasis of hormonal regulation is established, which does not impair growth. Changes caused by *AtCPK1-Ca* expression may be important under stressful conditions. Summarizing, one can suppose that AtCPK1 under stress conditions could modulate the mutual regulation of signaling systems with a strong bias to SAR through the modulation of primary metabolism (Figure 7).

## Figures and Tables

**Figure 1 ijms-24-13184-f001:**
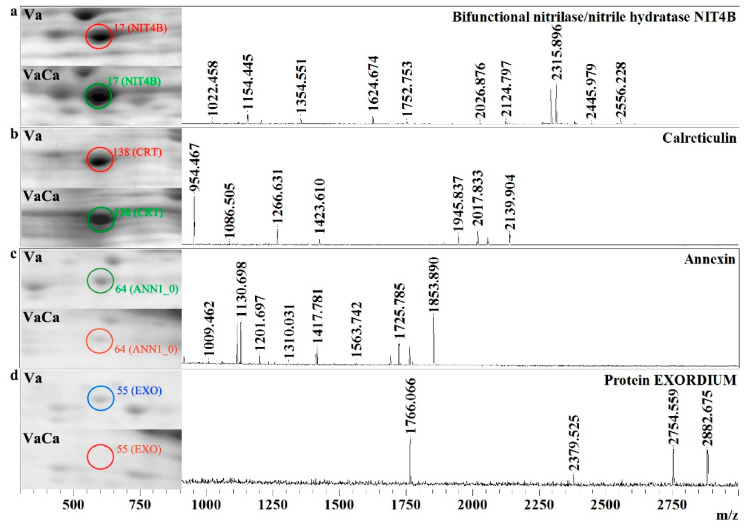
*AtCPK1*-Ca-triggered increased and decreased amounts of the proteins associated with signaling. Fragments of two-dimensional gels of protein fractions from the control (Va) and *AtCPK1*-Ca-transformed VaCa calli obtained in three separate experiments are presented in the left panel. Proteins were separated on gels by isoelectric mobility and by mass. Gels were scanned using a VersaDoc MP 4000 system with PDQuest 8.0.1 Advanced software, as described in Section 4.3. Protein expression is presented in Table 1 as the average total spot intensity in a group of three repeated gels. Mass spectra were acquired with a MALDI-TOF mass spectrometer and presented on the right panel. (**a**) bifunctional nitrilase/nitrilehydratase NIT4B; (**b**) calreticulin; (**c**) annexin; (**d**) protein EXORDIUM. Up-regulated proteins are marked with a green circle, down-regulated proteins are marked with a red circle, and newly arising proteins are marked with a blue circle.

**Figure 2 ijms-24-13184-f002:**
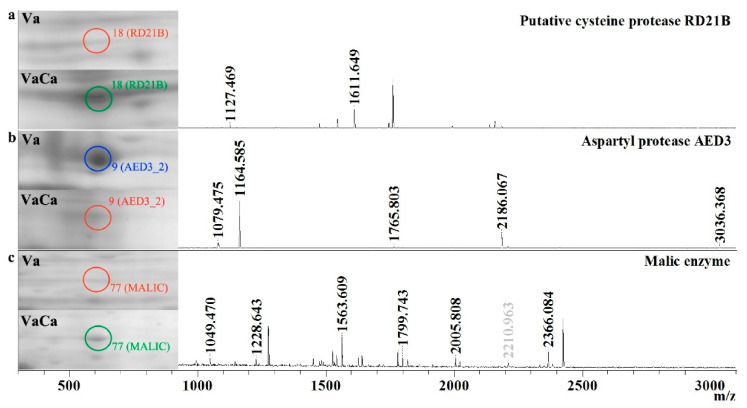
*AtCPK1*-Ca-triggered increased and decreased amounts of the enzymes involved in protein and carbohydrate metabolism. Fragments of two-dimensional gels of protein fractions from the control (Va) and *AtCPK1*-Ca-transformed VaCa calli, obtained in three separate experiments, are presented in the left panel. Proteins were divided on gels by isoelectric mobility and by mass. Gels were scanned using a VersaDoc MP 4000 system with PDQuest 8.0.1 Advanced software, as described in Section 4.3. Protein expression is presented in Table 1 as the average total spot intensity in a group of three repeated gels. Mass spectra were acquired with a MALDI-TOF mass spectrometer and presented on the right panel. (**a**) putative cysteine protease RD21B; (**b**) aspartyl protease AED3; (**c**) malic enzyme. Upregulated proteins are marked with a green circle, downregulated proteins are marked with a red circle, and newly arising proteins are marked with a blue circle.

**Figure 3 ijms-24-13184-f003:**
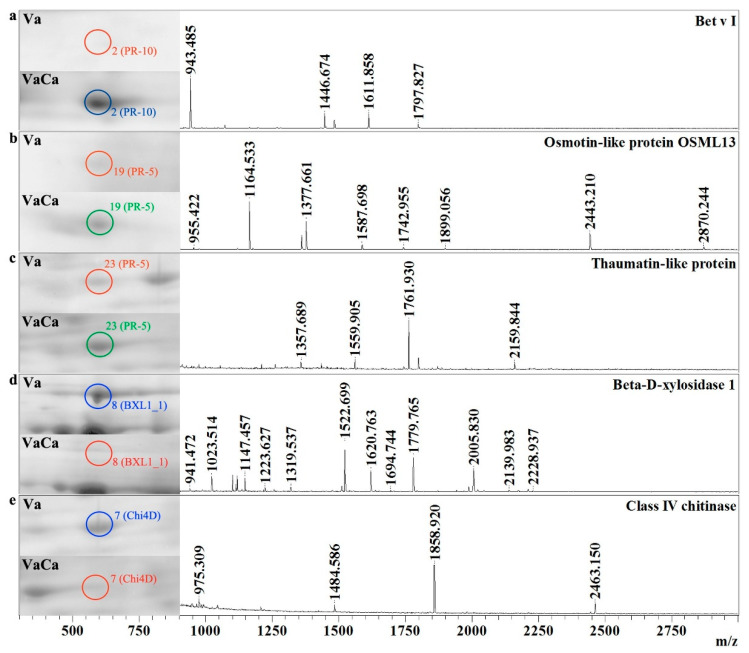
AtCPK1-Ca-triggered increased and decreased amounts of the PR-proteins. Fragments of two-dimensional gels of protein fractions from the control (Va) and AtCPK1-Ca-transformed VaCa calli obtained in three separate experiments are presented in the left panel. Proteins were separated on gels by isoelectric mobility and by mass. Gels were scanned using a VersaDoc MP 4000 system with PDQuest 8.0.1 Advanced software, as described in Section 4.3. Protein expression is presented in Table 1 as the average total spot intensity in a group of three repeated gels. Mass spectra were acquired with a MALDI-TOF mass spectrometer and presented on the right panel. (**a**) betvI/major latex protein; (**b**) osmotin-like protein 13; (**c**) thaumatin-like protein; (**d**) beta-D-xylosidase 1; (**e**) class IV chitinase. Up-regulated proteins are marked with a green circle, down-regulated proteins are marked with a red circle, and newly arising proteins are marked with a blue circle.

**Figure 4 ijms-24-13184-f004:**
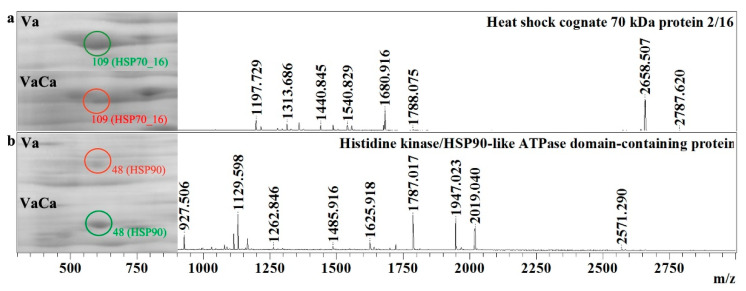
*AtCPK1*-Ca-triggered increased and decreased amounts of the HSP proteins. Fragments of two-dimensional gels of protein fractions from the control (Va) and *AtCPK1*-Ca-transformed VaCa calli obtained in three separate experiments are presented in the left panel. Protein was separated on gels by isoelectric mobility and by mass. Gels were scanned using a VersaDoc MP 4000 system with PDQuest 8.0.1 Advanced software, as described in Section 4.3. Protein expression is presented in Table 1 as the average total spot intensity in a group of three repeated gels. Mass spectra were acquired with a MALDI-TOF mass spectrometer and presented on the right panel. (**a**) heat shock cognate 70 kDa protein; (**b**) histidine kinase/HSP90-like ATPase domain-containing protein. Up-regulated proteins are marked with a green circle, down-regulated proteins are marked with a red circle, and newly arising proteins are marked with a blue circle.

**Figure 5 ijms-24-13184-f005:**
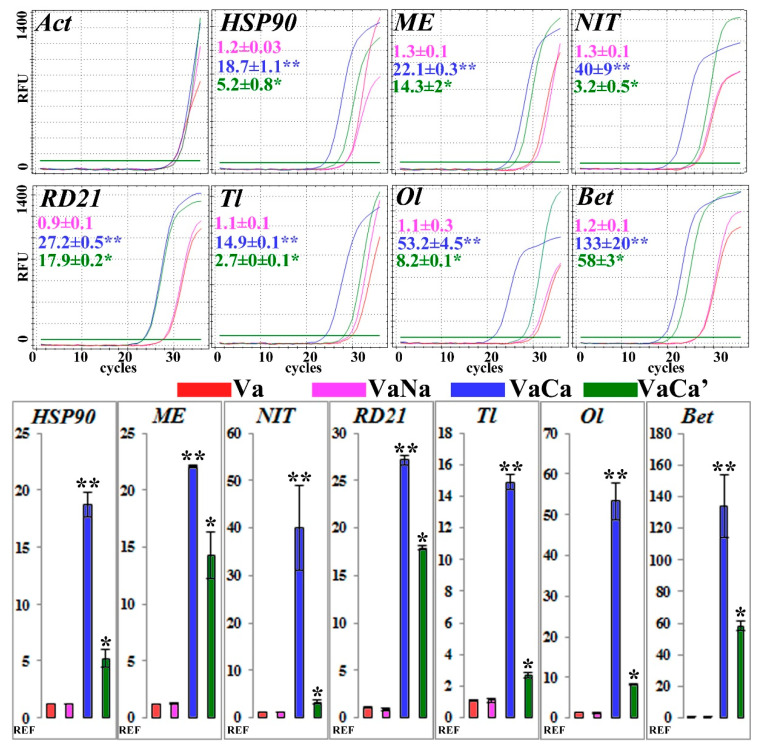
Representative results (**top**) of the qRT-PCR analysis (**bottom**) of some proteins. Relative changes in expression were compared to the control Va calli (red), calculated as 2^−ΔΔCt^, and presented in different colors: pink, VaNa; blue, VaCa; green, VaCa’. The VaNa line expressed the inactive form of *AtCPK1*, *AtCPK1-Na*. The VaCa’ line expressed the *AtCPK1*-*Ca* gene at a lower level than the VaCa line. Three independent RNA extractions were used for the analysis of the expression of genes of the most important *AtCPK1-Ca*-induced proteins: chaperones (*HSP90*); malic enzyme (*ME*); bifunctional nitrilase/nitrilehydratase (*NIT*); cysteine protease (*RD21*); PR5, thaumatin-like (*Tl*) and osmotin-like (*Ol*) proteins; *PR10*, betvI/major latex protein (*Bet*). The data are presented as the mean of three independent biological and three technical replicates with standard errors. The single asterisks indicate significantly different means of VaCa’ compared to Va and VaNa; twin asterisks indicate significantly different means of VaCa compared to Va, VaNa, and VaCa’ (*p* ≤ 0.05), Fisher’s LSD.

**Figure 6 ijms-24-13184-f006:**
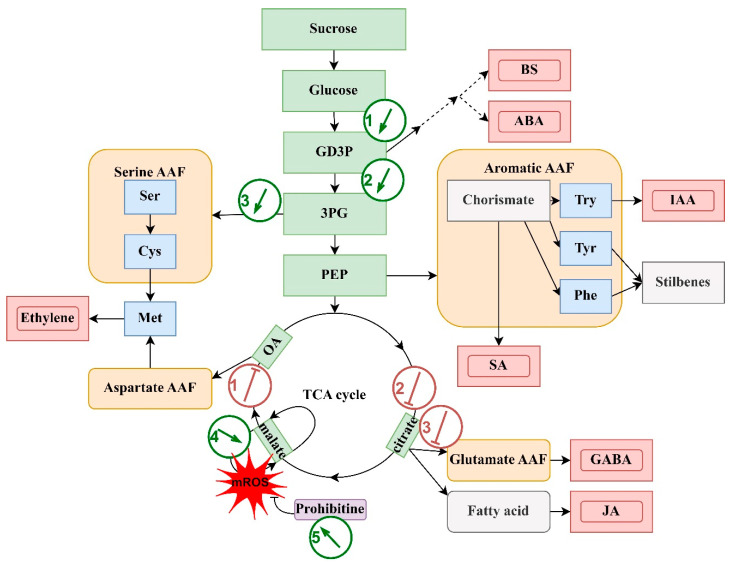
A simplified model suggesting the possible mechanisms of CPK action on primary metabolism. This model is based on a study by Trovato et al., 2021 [43]. Green boxes, sugar metabolism, including the TCA cycle; beige smoothed boxes, amino acid families (AAF); blue boxes, amino acids; red double boxes, signaling molecules; gray boxes, other derivatives; mROS, mitochondrial ROS; black arrows, direction of the pathways. The green arrows in the circles show the points of the AtCPK1-Ca-induced increase in enzyme abundance: 1 and 2, biosynthesis of penthose phosphate intermediates (with the participation of glyceraldehyde-3-phosphatedehydrogenase and phosphoglycerate kinase); 3, cysteine biosynthesis catalyzed by 3PG (cysteine synthase); 4 and 5, mitochondrial mROS production catalyzed by malic enzyme and reduction (by prohibitine), respectively. The red blunt lines in the circles show the points of the *AtCPK1-Ca*-induced decrease in abundance of enzymes: 1, biosynthesis of aspartate pool from oxaloacetate (by malate dehydrogenases and phosphoserine aminotransferase); the biosynthesis of glutamate from citrate (by isocitrate dehydrogenases, aconitate hydratases, glutamine synthetase, and glutamate dehydrogenase).

**Figure 7 ijms-24-13184-f007:**
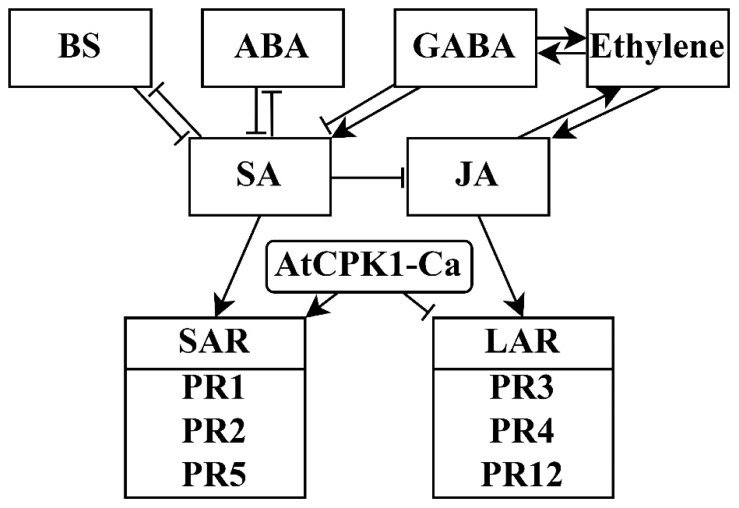
A simplified model suggesting possible mechanisms of CDPK action on the signaling molecules and PR-proteins. The presented model is based on a review by Enoki and Suzuki, 2016 [31]. The scheme shows crosstalk between brassinosteroids (BS), abscisic acid (ABA), γ-aminobutyric acid (GABA), ethylene, salicylic acid (SA), and jasmonic acids. As a result of the AtCPK1-Ca action, we assume the activation of systemic acquired resistance and inhibition of localized acquired resistance through PR proteins.

**Table 1 ijms-24-13184-t001:** Proteins identified in the control and *AtCPK1-Ca*-transformed cell cultures *V. amurensis*.

Spot Number/Short Name			Mean Value	Uniprot, *V.vinifera*/*A.thaliana/* TAIR ID, *A.thaliana*				
**Signaling**								
*Ethylene biosynthesis*								
101/NIT4B			*2	A0A438IX76_VITVI/P46011/At5g22300.1				
17/NIT4B_3			*8	A0A438IX97_VITVI/P46011/At5g22300.1				
147/METK4_1				A0A438D081_VITVI/P23686/At1g02500				
*Calcium signaling*								
138/CRT			*8	D7U2H8_VITVI/O04151/At1g56340.1				
138/CRT			*8	D7UA21_VITVI/Q38858/At1g09210.1				
64/ANN1_0			/5	A0A438FZR3_VITVI/Q9SYT0/At1g35720				
*Auxin signaling*								
105/DAO_1			*5	A0A438D270_VITVI/Q9XI75/At1g14130.1				
*Brassinosteroid signaling*								
55/EXO			/10	A0A438F8N5_VITVI/Q9ZPE7/At4g08950				
*MeJA signaling*								
35/TIFY10C				TI10C_ORYSI/Q9LMA8/At1g19180				
*ROS catabolic process*								
4/GST			*7	A0A438BWJ7_VITVI/Q9C8M3/At1g53680				
52/GSTU10_5			*2	A0A438CQC2_VITVI/Q9CA57/AT1G74590.1				
3/GSTF13_0			*7	A0A438F3E1_VITVI/Q96266-2/At2g47730				
140/CAT1_2				A0A438HMK9_VITVI/P25819/At4g35090				
153/CAT				A0A0F7G9V6_VITVI/P25819/At4g35090:				
123/MDAR3				A0A438JS91_VITVI/Q9LFA3/At3g52880				
10/PNC1_16			/9	A0A438C3M1_VITVI/Q9FLC0/At5g05340				
*Programmed cell death (PCD) signaling mechanism*								
28/EP1_2			*7	A0A438E5H3_VITVI/Q9ZVA4/AT1G78850.				
**DNA/RNA metabolic process**								
103/ADK2_1				A0A438GPM6_VITVI/Q9LZG0/At5g03300				
144/MPP				D7U090_VITVI/O04308/At3g16480				
127/NPK				A5B878_VITVI/P39207/At4g09320				
**Protein and amino acid synthesis**								
24/CYSK_6			*5	A0A438CWC5_VITVI/OASA1/AT4G14880.1				
119/SAHH_1				A0A438ISZ9_VITVI/O23255/At4g13940				
146/EIF4A3A				A0A438J912_VITVI/P41377/At1g54270				
95/METE				A5C7K7_VITVI/O50008/AT5G17920.1				
136/PDI_1				A0A438F8P6_VITVI/Q9XI01/At1g21750				
26/PDIA6_0				A0A438IMJ7_VITVI/O22263/At2g47470				
130/TEF1_1			/9	A0A438BNW3_VITVI/Q8GTY0/At5g60390.				
110/GS			/9	A0A0A0QQR7_9ROSI/Q56WN1/At5g37600				
71/KAR			/4	A5AGN5_VITVI/Q05758/At3g58610				
84/GDH1_1			/6	A0A438EGS1_VITVI/Q43314/At5g18170				
68/PSAT			/6	A0A438DBR8_VITVI/Q9SHP0/At2g17630				
**Protein and amino acid catabolism**								
18/RD21B			*9	A0A438ITG1_VITVI/ Q9FMH8/AT5G43060.1				
9/AED3_2			/10	A0A438G0C6_VITVI/O04496/At1g09750				
116/RPT1_1				A0A438EWK5_VITVI/Q9SSB5/At1g53750				
51/PAD1_1				A0A438DSU8_VITVI/O24616/At5g66140				
56/PAD				D7U7S6_VITVI/P42742/At3g60820				
118/MPPB				A0A438D4P9_VITVI/Q42290-2/At3g02090				
**Carbohydrate metabolic pathways**								
97/EPHX2_2			*3	A0A438JK35_VITVI/Q9SD45/**AT3g51000**				
96/FRK2			*3	A0A438DJZ1_VITVI/Q9M1B9/AT3G59480.1				
22/PDH			*3	F6I1P0_VITVI/Q38799/At5g50850				
102/SUCB_0			*4	A0A438F186_VITVI/O82662/At2g20420				
78/ADH1_8			*5	A0A438DAW3_VITVI/P06525/At1g77120				
43/FBA1_2			*5	A0A438EJU4_VITVI/Q9SJQ9/AT2G36460.1				
40/GAPDH			*7	F6GSG7_VITVI/Q9FX54/At1g13440.1				
47/PGM1_0			*5	A0A438D2Y0_VITVI/Q9M9K1/AT3G08590.1				
27/PGKY_0			*5	A0A438DH18_VITVI/Q9SAJ4/ AT1G79550.1				
77/NADP-ME			*8	A0A1Z2THL4_9ROSI/Q9XGZ0/At5g25880.1				
32/GAPC2_2			*9	A0A438DIR9_VITVI/P25858/At3g04120				
88/ACO			/9	D7T7Y3_VITVI/Q9SIB9/At2g05710.1				
89/ACO1_1			/9	A0A438EV70_VITVI/Q42560/At4G35830.1				
81/FDH1_0			/9	A0A438EFK8_VITVI/A0A1P8B9N1/At5g14780				
111/ICDH			/9	D7TQM9_VITVI/Q945K7/At5g03290				
65/FBA3_0			/7	A0A438JUA2_VITVI/Q9ZU52/At2g01140				
152/GME-1_0			/5	A0A438IH22_VITVI/Q93VR3/At5g28840				
86/ICDH-1			/5	A0A438DVA0_VITVI/Q9SRZ6/At1g65930				
115/UGPT			/7	F6I0H8_VITVI/P57751/At5g17310				
112/CMDH_1			/10	A0A438CMY8_VITVI/P57106/At5g43330				
112/CMDH_3			/10	A0A438KDL0_VITVI/P57106/At5g43330				
120/ALDH2B4_6			/2	A0A438EHU2_VITVI/Q9SU63/At3g48000				
79/MDH			/2	A0A1Z2THL9_9ROSI/Q9ZP06/At1g53240				
54/TPI			/4	A5BV65_VITVI/P48491/At3g55440				
75/ENO				F6HKH3_VITVI/P25696/				
**ATP synthesis**								
134/ATPB_1			*4	A0A438JUR7_VITVI/P83484/At5g08690.1				
135/ATPB_3			*3	F6GTT2_VITVI/P83484/At5g08690.1				
117/ATPA_1			/7	A0A654ICF0_9ROSI/F4IMB5/At2g07698				
139/VPP				A5B7R2_VITVI/Q9SZN1/At4g38510,				
108/VATA_3				A0A438E4U3_VITVI/O23654/At1g78900				
**Pathogenesis-Related Proteins**								
60/E13A_1			*5	A0A438I656_VITVI/F4J270/At3g57240.1				
37/BG3			*10	F6HLL8_VITVI/F4J270/At3g57240.1				
19/OS13_1 (PR-5)			*7	A0A438JJ78_VITVI/P50700/At4g11650.1				
2/BetvI (PR-10)			*10	D7SY76_VITVI/Q93VR4/At1g24020.1				
13/PR10.1			*9	Q9FS42_VITVI/Q93VR4/At1g24020.1				
11/PR10.2			*9	Q9FS43_VITVI/Q93VR4/At1g24020.1				
36/HGN1_0			*9	A0A438DWY2_VITVI/Q8VZJ2/At4g16260				
1/PR10.3			*10	B7SL50_VITVI/Q93VR4/At1g24020.1				
23/Tl3 (PR-5)			*5	Q7XAU7_VITVI/P50700/At4g11650.1				
20/EP3_15				A0A438HVS3_VITVI/Q9M2U5/At3g54420				
8/BXL1_0			/10	A0A438DWR0_VITVI/Q9FGY1/At5g49360				
70/Xyl2_0			/9	A0A438CN60_VITVI/Q9FLG1/At5g64570				
7/Chi4D			/10	Q7XAU6_VITVI/Q9M2U5/At3g54420				
63/E13ip5.23			/10	A0A438KGT6_VITVI/F4J270/At3g57240				
74/CEL1_4			/4	A0A438I4T3_VITVI/Q9SRX3/At1g02800				
**Chaperonins**								
48/HSP90			*7	E0CQ80_VITVI/Q9STX5/At4g24190.1				
143/CPN60-2_1			*5	A0A438BRG7_VITVI/P29197/At3g23990				
46/RUBB_0			*4	A0A438ELV8_VITVI/Q9LJE4/At3g13470.1				
107/HSP7M_1				A0A438FBI8_VITVI/Q9LDZ0/At5g09590				
109/HSP70_16			/8	A0A438JXP4_VITVI/P22953/At5g02500				
128/PCKR1_1			/9	A0A438E487_VITVI/P34790/At4g38740				
**Microtubule-based process**								
33/TUBA_1				A0A438C3Y7_VITVI/Q0WV25/At1g04820.1				
133/TUBA_3				A0A438HDT1_VITVI/Q0WV25/At1g04820.1				
141/TUBA5_3				A0A438EM73_VITVI/B9DHQ0/At5g19780.1				
142/TUB				F6HLZ6_VITVI/P29516/At5g23860				
98/ACT7_2				A0A438CNF0_VITVI/P53492/At5g09810.1				
99/ACT1_1				A0A438EPH9_VITVI/P53496/At3g12110.1				
**Membrane transport**								
67/VDAC1_1			*5	A0A438CTH2_VITVI/Q9SRH5/At3g01280				
67/PHB3_0			*5	A0A438BT27_VITVI/O04331/At5g40770				
**Other metabolic process**								
59/MO3_7			/9	A0A438ETS1_VITVI/O81816/At4g38540				
69/GDSL			/9	A0A438D1S8_VITVI/Q9LY84/At5g14450				
72/PE			/10	F6I0G4_VITVI/Q9LXD9/At5g09760				
125/FATB1_1				A0A438JQE3_VITVI/Q9SJE2/At1g08510				
61/SALR_0				A0A438C2P8_VITVI/Q94K30/At1g01800				
**Relative Expression Folds**								
**−10**	**−8**	**−5**	**−2**	**0**	**2**	**5**	**8**	**10**


Notes: The data presented as the mean of three biological repeats. The table shows proteins with statistically significant differences from the proteins of Va cells (*p* < 0.05, Student’s *t*-test). Fold changes in protein expression were assessed based on the mean protein spot intensity using PDQuest 8.0.1 Advanced software (see Section 4). Fold changes *10 or /10 indicate the absence of these proteins in the control Va or transgenic VaCa calli, respectively. Symbols * and / indicate increasing and decreasing, respectively.

## Data Availability

The mass spectrometry proteomics data has been deposited to the ProteomeXchange Consortium via the PRIDE [20] partner repository with the dataset identifier: project name: Function of constitutively active AtCPK1; project accession: PXD043507; project DOI: 10.6019/PXD043507.

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
