# Peer review of "Proteome-Level Investigation of Vitis amurensis Calli Transformed with a Constitutively Active, Ca2+-Independent Form of the Arabidopsis AtCPK1 Gene"

_ijms, 2023, doi:10.3390/ijms241713184_

Round 1

Reviewer 1 Report

1. Some important results or evidence were lacking to support the conclusion, and the story is not complete. The first one is how you designed the expression construct and how you prepared it; the second one is the evidence of successful transformation of this gene into the calli.

2. You mentioned that you did real-time PCR in this work, and you described it in the methods. However, it is hard to find related content in the result part.

3. what is your conclusion of this work? It was not described clearly.

Reviewer 2 Report

The manuscript by Veremeichik et al reports the proteome modifications induced by overexpressing a constitutively active form of the Arabidopsis calcium-dependent protein kinase CPK1 in Vitis amurensis calli. Using 2D-gels, they identified 79 proteins harboring increased or decreased abundance in CPK1 transgenic calli. For 6 of them, the authors could correlate the protein accumulation with increased transcription. This study is mainly descriptive and speculates on the putative roles of CPK1 in primary metabolism and PR-related defence.

Major comments:

1. Some spots contain 2 different proteins but the quantification is made on the size of the whole spot. So how could the authors assign a specific quantification to individual proteins in those spots ? For example, spot 134 contains TUBA_3 whose amount is constant and ATPB_1 which accumulates 4 fold.

2. The proteomic analysis has been performed by comparing one CPK1 transgenic line with an untransformed line. The data could be validated for at least few proteins, by western-blot with commercial antibodies (like HSP90, HSP70, PR proteins) and using the CPK1 inactive lines (CPK1-Na from Veremeichik et al 2017) as a control.

3. The result section is mostly a repetition of the data from table1, which could then be shortened. The manuscript would benefit from additional validations of their model by measuring SA, JA and ethylene in the lines Va, VaCa and CPK1-Na. Indeed, the model in Fig 6 places CPK1 independently of hormones whereas CPK1 was reported to trigger SA biosynthesis (Ref 19). Proteomic analysis with 2D-gel is not exhaustive and can miss some target proteins. Thus measuring hormone levels would confirm the upstream role of CPK1.

4. Throughout the manuscript, the authors use inhibition/activation of proteins/enzymes which is not correct: proteomics only detects amounts of proteins, not activity.

5. The numbers of proteins in each category listed in results l.119-123 don’t fit with the data in table 1. For example, in signaling, the numbers should be 17/9/3 instead of 19/9/5. Moreover, the supplementary table 1 is missing. The paragraph on CDPK activation mechanism should be revised. L.49, the N-terminal variable domain is rather involved in substrate specificity than activation per se. L.53, the junction fragment blocks kinase activity when [Ca2+] is low/in resting conditions, not released. The authors could cite more specific reviews on this aspect (Liese and Romeis, 2013, Biochimica et Biophysica Acta 1833: 1582–1589; or Yip Delormel and Boudsocq, 2019, New Phytol 224: 585–604).

Minor comments:

1. The sentence l.82-84 is duplicated and should be deleted.

2. L.77 and 78, “constitutive” should be replaced by “conserved”. L.76 and 79, “CAM-LD” (which refers to the C-terminal CAM-like domain with 4 EF-hands) should be replaced by “CAM-BD” (which refers to the small domain where CAM/CAM-LD bind).

3. In the table p.9, for spot 36/HGN1_0, “*8” should be “/8”.

4. From the methods l.423-426, it is not clear whether the MS analysis was performed on the 3 biological replicates or only one but with several technical replicates.

5. In the results l.124-125, the sentence should be rephrased: qRT-PCR data correlate with accumulation of some proteins in VaCa cells, which may result from this induced transcription. But it’s not a confirmation of proteomics data. For example, PR10 is induced only x2.7 in qPCR but the protein accumulates x10. So additional mechanisms are required to explain the increased abundance of PR10.

6. The model in Fig 5 should include the enzymes involved to correlate with the proteomic data.

7. Some citations are wrong so references should be checked again. For example, l.373, Ref 43 should be 44 I guess; l.454 ref 42 should be 43; l.461, Ref 43 is wrong.

The manuscript requires thorough editing to correct English and several sentences are misleading: l.47; l.61; l.68-70; l.74-75; l.131; l.134; l.159-160; l.171; l.236-238; l.396-397; l.437-440; l.444-445; l.463-464. For example, in figure legend of Fig1/2/3/4, the sentence “Protein was divided on gels into two isoforms by isoelectric mobility and by mass” is not clear and should be rephrased.

Round 2

Reviewer 1 Report

1. If Fig.5 can be presented with histogram, it may be better and more clear.

2. The conclusion part can be simplified. There is no need to include so much information from previous studies. Directly pointing out what you found is more important and straightforward.

Reviewer 2 Report

The authors included new qRT-PCR data with additional controls to strengthen the proteomic results. However, they couldn’t reveal any modifications in hormone measurement. Moreover, several previous comments are still unresolved.

Major comments:

1. The methods still don’t explain how 2-3 proteins in the same spot could be quantified individually. The quantification has been made on the spot size of the 2D-gel. If the authors further used MS data for quantification, they need to explain the process in the methods, and especially the normalization. The excel file with all the identifications and quantifications must be provided as supplementary data.

2. The measurement of hormones didn’t reveal any differences and has been performed in an independent system (tobacco lines). Thus this is not informative and should be deleted. Without further validation, the discussion is still highly speculative. For example, the authors speculate on the increase of ethylene while the rate-limiting enzyme of ethylene biosynthesis ACS (which accumulates upon stress) has not been detected.

3. The introduction still requires several modifications: l.57, “prevalent” should be “studied”; l.61, “with PK” should be “with PK domain”; l.64, “released” should be “low”; l.77 “latent” should be “inactive”; l.86, “CAM/CAM-LD binding” must be deleted (it introduces confusion with the CAM-LD = calmodulin-like domain corresponding to the 4 EF-hands). And the paragraph describing KJM23 and KJM4 mutants is still redundant from l.73 to l.90.

4. There are still mistakes in the numbers of proteins identified: l. 121, only 113 proteins are presented in table 1; l. 122, 38 proteins (and not 44) are downregulated; l. 126, in signaling, the numbers should be 17/9/3 instead of 16/9/3.

5. The authors simplified table 1 but the result part is still a long description of the table. In particular, the proteins which don’t show any different accumulation in AtCPK1-Ca line don’t need to be described in details. This will better highlight the proteins whose amounts change in AtCPK1-Ca line. The description l. 206-208 and l. 261-273 don’t fit the new data. L. 292, “seven” should be “eight”. L. 241-242, the sentence needs to be rephrased. L. 242/246, “and others” is not informative.

Minor comments:

1. In supplementary table S1, for spot 36/HGN1_0, “*8” should be “/8”.

2. There are still some editing required: l. 139, “this protein”; l.142-144, it should be only one sentence I guess; l. 149 “methionine”; l. 283 and l. 306, “Proteins were”; l. 293-294, the sentence is incomplete.

3. In the legend of Fig 2/3/4, the colors are inverted: “Up-regulated proteins are marked with a green circle, down-regulated proteins are marked with a red circle”. The title would be clearer with : “AtCPK1-Ca-triggered increased and decreased amounts of the proteins/enzymes …”. L. 225, the sentence has still not been modified.

4. Supplementary figure S1 and S2 are copies of supplementary Fig 1 from Ref 40 and Fig 1 from Ref 16, respectively. I’m not sure that the authors are allowed to reproduce the figures here but they should cite those articles.

5. Ref 16 should be in 2017. L. 443, Ref 39 should be 38.

6. For Fig 5, the legend doesn’t explain the difference between 1 and 2 asterisks.

7. The authors modified the qPCR design (new lines, independent growth) but the methods have not been modified. L. 530, the authors should cite all the databases and programs they used.

English has been improved but some sentences still need to be rephrased.
